# The Biological Functions and Intestinal Inflammation Regulation of IL-21 in Grass Carp (*Ctenopharyngodon idella*) during Infection with *Aeromonas hydrophila*

**DOI:** 10.3390/cells12182276

**Published:** 2023-09-14

**Authors:** Gaoliang Yuan, Weihua Zhao, Yanwei Zhang, Zhao Jia, Kangyong Chen, Junya Wang, Hao Feng, Jun Zou

**Affiliations:** 1Key Laboratory of Exploration and Utilization of Aquatic Genetic Resources, Ministry of Education, Shanghai Ocean University, Shanghai 201306, China; d210100030@st.shou.edu.cn (G.Y.); m220100311@st.shou.edu.cn (W.Z.); d210100021@st.shou.edu.cn (Y.Z.); d210100015@st.shou.edu.cn (Z.J.); d210100017@st.shou.edu.cn (K.C.); jywang@shou.edu.cn (J.W.); 2International Research Center for Marine Biosciences, Ministry of Science and Technology, Shanghai Ocean University, Shanghai 201306, China; 3National Demonstration Center for Experimental Fisheries Science Education, Shanghai Ocean University, Shanghai 201306, China; 4State Key Laboratory of Developmental Biology of Freshwater Fish, College of Life Science, Hunan Normal University, Changsha 410081, China; fenghao@hunnu.edu.cn; 5Laboratory for Marine Biology and Biotechnology, Qingdao National Laboratory for Marine Science and Technology, Qingdao 266200, China

**Keywords:** interleukin 21, T cells, intestinal inflammation, bacterial infection, fish

## Abstract

Interleukin (IL) 21 is a pleiotropic cytokine that plays an important role in regulating innate and adaptive immune responses. In fish, the biological functions and cell source of IL-21 remain largely unknown. In this study, we performed qRT-PCR, Western blotting and immunofluorescent microscopy to examine the expression of IL-21 at the mRNA and protein levels. We found that *il21* expression was induced in the primary head kidney leukocytes of grass carp (*Ctenopharyngodon idella*) by heat-inactivated *Aeromonas hydrophila* (*A. hydrophila*) and LPS and in tissues after infection with *A. hydrophila*. Recombinant IL-21 protein produced in the CHO-S cells was effective in elevating the expression of antibacterial genes, including *β-defensin* and *lysozyme*, and, interestingly, inhibited the NF-κB signaling pathway. Furthermore, we investigated the response of the IL-21 expressing cells to *A. hydrophila* infection. Immunofluorescent assay showed that IL-21 protein was detected in the CD3γ/δ T cells and was markedly accumulated in the anterior, middle and posterior intestine. Collectively, the results indicate that IL-21 plays an important role in regulating the intestinal inflammation induced by bacterial infection in grass carp.

## 1. Introduction

Interleukin 21 (IL-21) belongs to the gamma chain (γc) family of type I cytokines and possesses a four α-helical structure. It is produced mainly by immune cells, including natural killer T (NKT) cells and activated T cells [1,2,3,4]. IL-21 activates a heterodimeric receptor complex consisting of IL-21R and γc receptor to exert diverse functions in the regulation of innate and adaptive immune responses [5,6,7]. Similar to other members of the γc cytokine family such as IL-2, IL-4, IL-7, IL-9 and IL-15, IL-21 initiates multiple signaling pathways, including JAK/STAT and MAPK signaling pathways. IL-21 acts on macrophages, T cells, B cells, dendritic cells (DC), natural killer (NK) cells and non-immune cells. For instance, IL-21 enhances phagocytosis of macrophages and polarizes macrophages from M1 to M2 phenotype [8,9]. Furthermore, it enhances survival and proliferation of NKT cells and affects NKT cell granularity [10]. Other studies have shown that IL-21 promotes the expansion, differentiation and proliferation of T cells and drives differentiation and maturation of B cells in the germinal centers to produce specific antibodies [11,12]. Interestingly, IL-21 inhibits activation and maturation of DCs induced by lipopolysaccride (LPS) [13].

In 2005, the first fish *il21* gene was sequenced in the Japanese puffer *Fugu rubripesi* [14]. Subsequently, *il21* genes have been reported in spotted green pufferfish (*Tetraodon nigroviridis*) [15], zebrafish [16] (*Danio rerio*), rainbow trout (*Oncorhynchus mykiss*) [17], Japanese flounder (*Paralichthys olivaceus*) [18], grass carp (*Ctenopharyngodon idella*) [19,20,21] and snakehead (*Channa argus*) [22]. Similar to mammalian IL-21 counterparts, teleost IL-21 homologs consist of four α-helices and two pairs of conserved disulfide bonds [14,17,23]. However, the genomic organization of teleost *il21* differs from that in mammals. Most teleost *il21* genes have six exons and five introns whilst mammalian *il21* genes have five exons and four introns [14,17,23]. It must be noted that the fugu *il21* gene has the same genomic organization as mammalian homologs. Two IL-21 receptors are present in teleosts and both possess a conserved extracellular cytokine-binding homology domain (CHD), a transmembrane domain and an intracellular domain [23].

The expression of fish *il21* has been studied mostly at the transcriptional level. It has been shown that fish *il21* genes are constitutively expressed in most tissues but more abundantly in the head kidney, gills and intestine [14,15,17,18,20,21,22]. The *il21* expression can be modulated by immune stimuli and pathogenic infections. For instance, grass carp and rainbow trout *il21* genes have been shown to be upregulated in the primary leukocytes (HKLs) by phytohemagglutinin (PHA) and phorbol myristate acetate (PMA), known to activate T cells, suggesting that *il21* could be produced in activated T cells [17,21]. In grass carp, splicing variants of *il21* were identified and could be induced by LPS and polyinosinic–polycytidylic acid (poly(I:C)) in the head kidney leukocytes (HKLs) [20]. Moreover, remarkable increases of *il21* expression have been detected in the immune tissues of fish infected with bacterial pathogens such as *Yersinia ruckeri* (*Y. ruckeri*), *Edwardsiella tarda* (*E. tarda*), *Nocardia seriolae* (*N. seriolae*), *Aeromonas schubertii* (*A. schubertii*) and *Flavobacterium columnare* (*F. columnare*) [17,18,21,22]. It has also been shown that *il21* is induced by viral hemorrhagic septicemia virus in the kidney of rainbow trout at the early stage of infection [17]. In our previous studies, we showed that the IL-21 protein was detected in the primary kidney leukocytes stimulated with PHA and in tissues of fish infected with *F. columnare* using a monoclonal antibody [21]. This study aims to investigate whether IL-21 is produced in T cells and, if so, whether the IL-21 producing T cells can be modulated after infection with bacterial pathogens.

Fish are lower vertebrates and have both innate and adaptive immune systems [24]. Like mammals, fish possess a complex of cytokine networks for the coordination of immune responses to infections [25]. Grass carp is a herbivorous freshwater fish species with economic importance in China. It lacks a developed stomach but rather has a unique widening esophagus termed an intestinal bulb. The unusually long intestine can be subdivided into anterior, middle and posterior segments. It is pivotal for food digestion and the absorption of nutrients [26] and form one of the major mucosal sites for pathogen invasion [27]. It has been reported that enteritis caused by bacterial infections results in high mortality and huge economic losses [28,29]. For instance, *Aeromonas hydrophila* (*A. hydrophila*) induced enteritis is a common disease affecting the grass carp farming industry [29]. In addition to grass carp, *A. hydrophila* infects a wide range of other fish species, including common carp (*Cyprinus carpio*), blunt snout bream (*Megalobrama amblycephala*) and Chinese sea bass (*Lateolabrax maculatus*), causing intestinal damage and malfunction [30,31,32]. Thus, maintaining good intestinal health is vital to protect the body against pathogen infections.

In this study, we examined the expression patterns of IL-21 in fish infected with *A. hydrophila* at the transcript and protein levels. Using the monoclonal antibodies previously generated, the IL-21 producing cells were analyzed in different segments of the intestine by immunofluorescent microscopy. In addition, we determined the bioactivity of IL-21 in modulating antibacterial and inflammatory response. Our findings shed light on the functions of IL-21 in regulating the intestinal immunity of bony fish during bacterial infection.

## 2. Materials and Methods

### 2.1. Experimental Fish, Cells and Bacteria

Grass carp (15 ± 2 g) were obtained from Shanghai Binhai Base, Shanghai Ocean University, and acclimatized in a circulating freshwater aquarium at 28 ± 1 °C for at least two weeks before experiments. Fish were anesthetized with MS-222 (100 mg/L) prior to experimental procedures. All experiments were conducted under the national regulations on laboratory animals of China and approved by the laboratory animal ethics committee of Shanghai Ocean University (SHOU-DW-2021-027).

*Epithelioma papulosum cyprini* (EPC) cells (ATCC, Manassas, VA, USA) were maintained in Dulbecco’s Modified Eagle’s Medium (DMEM) supplemented with 10% fetal bovine serum (FBS, Gibco, Waltham, MA, USA) and 1% penicillin–streptomycin solution (Pen/Strep) at 28 °C in a 5% CO_2_ incubator [33]. CHO-S cells (ATCC, USA) were grown in ExpiCHO expression medium at 37 °C in an 8% CO_2_ shaker. *A. hydrophila* was provided by the National Pathogen Collection Center for Aquatic Animals, Shanghai Ocean University, China. *A. hydrophila* was grown at 28 °C in Luria–Bertani (LB) medium in a shaker (180 rpm) until absorbance at optical density (OD) 600 nm reached 0.6.

### 2.2. Plasmids and Reagents

The open reading frame (ORF) of grass carp *il21* (GenBank accession number: KP226585.1) was cloned into pcDNA3.1(+) vector with a 6-histidine tag at the C-terminus of IL-21. The plasmid was termed pcDNA3.1-IL21-His. *Il21* promoters (pro-1, pro-2 and pro-3) were amplified by PCR and cloned into pGL4.0 vector. All of the primers used in plasmid construction are listed in Appendix A. The NF-κB luciferase reporter plasmid and renilla luciferase control plasmid (pRL-TK) were purchased from Promega (Madison, WI, USA).

The main reagents included TRIzolTM Reagent, Hifair^®^ II 1st Strand cDNA Synthesis SuperMix and Hieff UNICON^®^ Power qPCR SYBR Green Master Mix (Yeasen, Shanghai, China). ExpiFectamine™ CHO Transfection Kit (Gibco, Grand Island, NY, USA), jetOPTIMUS Transfection Reagent (Polyplus, Illkirch, France), lipopolysaccharide (LPS, Sigma-Aldrich, St. Louis, MO, USA). Optimal Cutting Temperature Compound (OCT, Solarbio, Beijing, China), 4% paraformaldehyde (PFA, Solarbio, China), rabbit polyclonal antibody against β-actin (Huabio, Hangzhou, China), mouse monoclonal antibody against His-tag (Abmart, Shanghai, China), goat α-rabbit or α-mouse IgG secondary antibody (LI-COR, Lincoln, NE, USA), mouse monoclonal antibodies (GC9-IL21-FITC, IgG1) or CD3γ/δ (GC38T-Cy5, IgG2b) were generated and characterized in our laboratory [21,34]. FITC labeled mouse IgG1 and IgG2b (isotype antibodies) were purchased from Huabio, Hongzhou, China.

### 2.3. Bacterial Challenge

*A. hydrophila* were prepared as previously described [35], and were resuspended in sterile PBS at 1 × 10^7^ CFU/mL. Thirty grass carp (15 ± 2 g) were randomly placed into two tanks, each containing 15 fish. Fish were intraperitoneally (i.p.) injected with 100 μL of 1 × 10^7^ CFU/mL *A. hydrophila* or PBS (control). After 24, 48 and 72 h, peripheral blood leukocytes (PBLs) and immune tissues (25 ± 5 mg), including head kidney, gills, spleen, anterior intestine, middle intestine and posterior intestine were sampled from 4 fish of each group at 24, 48 and 72 h after infection for extraction of total RNA and proteins. In addition, three different intestine segments were sampled at 72 h post-infection and fixed in 4% paraformaldehyde (PFA) for cryosection.

### 2.4. Hematoxylin and Eosin (H&E) Staining of Grass Carp Intestine

Grass carp (15 ± 2 g) were i.p. injected with 100 μL *A. hydrophila* (1 × 10^7^ CFU/mL) or PBS. After 24, 48 and 72 h, anterior, middle and posterior intestine segments were collected and fixed in 4% neutral buffered paraformaldehyde solution for 24 h. The tissue samples were then embedded in paraffin and sectioned at a thickness of 4 μm. Finally, the slides were stained with hematoxylin and eosin (H&E) and photographed under an optical microscope [36].

### 2.5. Expression Analysis of il21 in the Primary HKLs after Stimulation with Inactivated A. hydrophila and LPS

Heat-inactivated *A. hydrophila* were prepared as previously described [35]. In brief, when OD_600_ of *A. hydrophila* culture reached 0.6, 1 mL of bacteria were collected and washed three times with sterile PBS. Bacterial cells were resuspended with 1 mL sterile PBS and placed in a water bath at 100 °C for 30 min. The inactivated *A. hydrophila* were stored at 4 °C. Grass carp primary HKLs were prepared as previously described [37] and seeded in 12-well plates (1 × 10^7^ cells/mL). After 6 h cultivation, HKLs were stimulated with inactivated 100 μL *A. hydrophila* (1 × 10^7^ CFU/mL) or LPS (75 μg/mL). Control cells were treated with PBS. Cells were collected at 12 h and 24 h for qRT-PCR and Western blotting analysis. In addition, HKLs were stimulated with inactivated *A. hydrophila*, LPS or PBS for 12 h and 24 h, and fixed with 4% PFA for immunofluorescent microscopy.

### 2.6. RNA Extraction, cDNA Synthesis and qRT-PCR

Total RNA was extracted from tissues (25 ± 5 mg) and cell samples using TRIzol^TM^ reagent. The first-strand cDNA was synthesized using the Hifair^®^ II 1st Strand cDNA Synthesis SuperMix and stored at −80 °C. qRT-PCR was performed using Hieff UNICON^®^ Power qPCR SYBR Green Master Mix and the LightCycler 480 Instrument (Roche, Basel, Switzerland). qRT-PCR was performed using the following conditions: 1 cycle of 95 °C for 30 s, 40 cycles of 95 °C for 5 s, 62 °C for 30 s, 72 °C for 10 s, followed by 1 cycle of 95 °C for 10 s, 65 °C for 60 s, 97 °C for 1 s. qRT-PCR was set up as follows: 5 μL SYBR^®^ Green PreMix Ex Taq™ II (Yeasen, Shanghai, China), 1 μL cDNA template, 0.2 μL forward primer (10 μM), 0.2 μL reverse primer (10 μM), and 3.6 μL H_2_O. The primers used for qRT-PCR assay in this study are listed in Appendix A. The elongation factor-1α (*ef1α*) gene was used as reference gene for normalization of gene expression. Data were analyzed using the 2^−△△Ct^ method [38].

### 2.7. Western Blotting

Proteins used for Western blotting were extracted from the intermediate protein layer between the aqueous and organic phase in RNA extraction using TRIzol^TM^ reagent. In brief, the protein pellet (white) was collected between the aqueous and organic phases, briefly dried and resuspended in 300 μL absolute ethanol. The protein solution was placed at room temperature for 2–3 min before centrifugation at 2000× *g* at 4 °C for 5 min. The upper organic phase was carefully transferred to a fresh centrifuge tube, and gently mixed with 1.5 mL isopropyl alcohol, followed by incubation at room temperature for 10 min. Protein pellet was obtained by centrifugation at 12,000× *g* at 4 °C for 10 min, washed three times with 0.3 M guanidine hydrochloride, resuspended in 2 mL ethanol for 20 min, and dissolved in 1% SDS (sodium dodecyl sulfate) at 50 °C. The protein solution was centrifuged at 12,000× *g* at 4 °C for 10 min and stored at −20 °C before use.

Western blotting analysis was performed. Briefly, the protein samples were dissolved in lysis buffer at 4 °C for 30 min. Subsequently, protein solution was mixed with 5×SDS protein loading buffer and boiled for 10 min. Protein solution was separated on a 12% SDS-PAGE gel and transferred onto a polyvinylidene difluoride (PVDF) membrane (Millipore, Billerica, MA, USA) using a semidry transfer unit (Bio-Rad, Hercules, CA, USA). The membrane was blocked with TBST buffer containing 5% skimmed milk for 1 h, followed by incubation with primary antibodies (1:10,000, *v*/*v*, 5 μg/mL) at 4 °C overnight. After washing with TBST three times, the membrane was incubated with secondary antibodies (1:10,000, *v*/*v*, 0.5 μg/mL) for 1 h and washed extensively with TBST. The membrane was photographed using the Odyssey CLx Imaging System (LI-COR, Lincoln, NE, USA).

### 2.8. Immunofluorescent Microscopy

Freshly prepared primary HKLs were washed with PBS, cytospinned onto a coverslip, and fixed with 4% PFA for 20 min. Cells were then permeabilized with PBS containing 0.3% Triton X-100 for 15 min, and blocked in PBS containing 5% bovine serum albumin (BSA) for 1 h. Coverslips were overlaid with the FITC-labeled GC9-IL21 mAb (1:200, *v*/*v*, 9 μg/mL) in darkness at 4 °C overnight and incubated with 1 μg/mL 4,6-diamidino-2-phenylindole (DAPI, Beyotime, Shanghai, China) in darkness for 10 min. The coverslips were observed under a Leica confocal microscope (Leica SP8, Wetzlar, Germany) and photographed.

Immunofluorescent microscopic analysis of tissues was carried out as previously described [39]. In brief, the intestine segments of grass carp were fixed with 4% PFA, and gradually dehydrated with sucrose solutions of different concentrations (5%, 10%, 15%, 20% and 30%). Tissues were embedded with OCT at −80 °C and sectioned into 8 μm. The cryosectioned tissue on glass slides was first placed in a staining container at room temperature for 30 min and washed with PBS to remove OCT. The slides were permeabilized with PBS containing 0.5% Triton X-100 (Sangon Biotech, Shanghai, China) for 15 min and washed three times with PBS. Antigen retrieval was performed by placement of the slides in 10 mM sodium citrate solution and subsequent incubation at 80 °C for 15 min. After washing, the slides were blocked with 5% BSA at room temperature for 1 h and incubated with GC9-IL21-FITC mAb (1:200, *v*/*v*, 9 μg/mL) and GC38T-Cy5 mAb (1:200, *v*/*v*, 6.45 μg/mL) at 4 °C overnight. After washing, slides were stained with DAPI and photographed under a Leica confocal microscope.

### 2.9. Production and Bioactivity Testing of Recombinant IL-21

Recombinant IL-21 (rIL-21) protein was expressed in CHO-S cells and purified. In brief, CHO-S cells were cultured in a 37 °C shaker (125 rpm) supplemented with 8% CO_2_ until cell numbers reached 6 × 10^6^ cells/mL. Twenty-five µg pcDNA3.1-IL21-His plasmids were diluted with 1 mL OptiPRO medium containing 80 µL ExpiFectamine™ CHO Reagent and 920 µL OptiPRO medium. After a 5 min incubation, the mixture was added into the flask containing CHO-S cells (6 × 10^6^ cells/mL), and after 18 h, 150 µL ExpiFectamine CHO Enhancer and 6 mL ExpiFectamine CHO Feed were added. Five days later, the same volume of ExpiFectamine CHO Enhancer and ExpiFectamine CHO Feed were added again. At day 8 post-transfection, culture media containing recombinant proteins were collected and applied for protein purification. The purity and size of proteins were verified by SDS-PAGE and Western blotting, which are described below. Purified proteins were aliquoted and stored at −80 °C until use.

To determine the bioactivity of the rIL-21 protein, primary HKLs of the grass carp were prepared according to our previous study [37]. The cells were seeded into 12-well plates and cultured in DMEM medium containing 10% FBS and 1% Pen/Strep at 28 °C in a 5% CO_2_ incubator. The rIL-21 protein was added into the cells at final concentrations of 5, 50, and 100 ng/mL. Control cells were treated with PBS. After 24 h, the cells were collected for qRT-PCR analysis.

### 2.10. Luciferase Promoter Reporter Assay

EPC cells have been previously shown to have high efficiencies for plasmid transfection and commonly used for in vitro studies [40]. Therefore, EPC cells were chosen for luciferase promoter reporter assay. EPC cells were seeded in 24-well plates until confluence reached 80%. The *il21* promoter plasmids or pGL4.0 (500 ng) were co-transfected with pRL-TK (50 ng) using jetOPTIMUS Transfection Reagent. The transfected cells were treated with inactivated *A. hydrophila*, LPS or PBS and harvested at 12 h and 24 h for promoter activity analysis. Alternatively, EPC cells were co-transfected with NF-κB reporter plasmid (500 ng) and pRL-TK (50 ng) using jetOPTIMUS Transfection Reagent. After 24 h, the cells were incubated with PBS or different concentrations of rIL-21 protein (5, 50 and 100 ng/mL) for 24 h. The cells were collected and lysed with passive lysis buffer at 4 °C for 30 min, and the luciferase activity was measured following the manufacturer’s protocol.

### 2.11. Statistical Analysis

Data were statistically analyzed using the SPSS 22.0 software. The statistical significance of biotesting of rIL-21 protein was analyzed for one-way ANOVA and mean comparisons between tested groups were conducted using Duncan’s new multiple range test (DMRT). The statistical significance of other data between the treatment and control groups was determined by Student’s *t*-test. *p* < 0.05 and *p* < 0.01 were considered significantly different.

## 3. Results

### 3.1. Pathological Changes in Different Segments of Intestine after Infection with A. hydrophila

Grass carp are herbivorous fish and are devoid of a well-developed stomach, but instead possess a unique widening esophagus termed intestine bulb [41]. In this study, we examined the histological changes of the anterior, middle and posterior segments of the intestine after infection with *A. hydrophila*. As shown in Figure 1A, H&E staining shows that the anterior, middle and posterior intestine of the PBS treated group display normal structures with densely arranged microvilli and the presence of fewer goblet cells and leukocytes. In contrast, *A. hydrophila* infection resulted in damage to the microvilli, massive infiltration of leukocytes and increases of goblet cells in all three intestinal segments (Figure 1B).

### 3.2. Tissue Expression Patterns of il21 after A. hydrophila Infection

We examined the expression patterns of *il21* in the gills, kidney, spleen, PBLs and different segments of the intestine upon *A. hydrophila* infection. In the gills, *il21* was significantly upregulated after infection at 24, 48 and 72 h (Figure 2A,B). Similar patterns of *il21* upregulation were observed in PBLs (Figure 2G,H). Relative to control group, *il21* expression in the kidney and spleen increased at 24 h, and then returned to the basal levels at 72 h (Figure 2C–F). Interestingly, the expression levels of *il21* in different intestinal segments varied significantly. *Il21* was induced in the anterior intestine at 24, 48 and 72 h post-infection (Figure 3A,B). In the middle intestine, *il21* was not affected at 24 and 48 h, while it was induced at 72 h upon *A. hydrophila* infection (Figure 3C,D). Likewise, in the posterior intestine, no change of *il21* expression was detected at 24 h while upregulation was observed at 72 h (Figure 3E,F). Curiously, *il21* was downregulated at 48 h. In general, the expression of pro-inflammatory cytokines such as *il1β* and *tnfα* was upregulated in all three segments of intestine except for *il1β* in the posterior intestine at 24 h and *tnfα* in the anterior intestine at 24 and 48 h post-infection (Figure 3G–L).

### 3.3. Analysis of IL-21 Producing CD3γ/δ T Cells in Different Segments of Intestine after A. hydrophila Infection

In our previous study, we showed that *A. hydrophila* infection resulted in mild enteritis in grass carp [39] and that IL-21^+^ and CD3γ/δ^+^ cells are present in the intestine [21,34,42]. Here, we sought to determine whether IL-21 is produced by CD3γ/δ cells and, if so, whether the IL-21 producing CD3γ/δ cells are involved in enteritis. As shown in Figure 4, the IL-21^+^ cells markedly increased in all three intestinal segments of fish infected with *A. hydrophila* after 72 h, which is consistent with the results obtained by qRT-PCR and Western blotting (Figure 3). Of note, the IL-21^+^ cells were mainly located in the submucosa. Interestingly, the CD3γ/δ^+^ cells were also detected and induced in the submucosal layer of infected fish. We observed that fluorescent staining of IL-21 and CD3γ/δ were mostly overlapped, indicating that the CD3γ/δ^+^ cells were activated to produce IL-21 by *A. hydrophila*. In addition, isotype antibodies (IgG1-FITC and IgG2b-Cy5) were used as controls to exclude false positive cells (Appendix A). Taken together, our results indicate that IL-21 was produced by CD3γ/δ^+^ cells and that IL-21/CD3γ/δ^+^ cells were markedly upregulated in the anterior, middle and posterior intestine after infection with *A. hydrophila*.

### 3.4. Expression Analysis of il21 in the Primary HKLs after Stimulation with LPS and Inactivated A. hydrophila

Head kidney (HK) is the hemopoietic organ producing immune cells in fish. To examine IL-21 response to stimulation with LPS and inactivated *A. hydrophila*, primary HKLs were stimulated with LPS or heat-inactivated *A. hydrophila*, and the expression of IL-21 was analyzed by qRT-PCR and confocal microscopy. We found that *il21* mRNA levels increased after stimulation with LPS and inactivated *A. hydrophila* at 12 (*p* < 0.05) and 24 h (*p* < 0.05 or *p* < 0.01) (Figure 5A,B). It appears that heat-inactivated *A. hydrophila* were more effective than LPS in inducing the expression of *il21* (Figure 5B). Consistent with these results, the IL-21^+^ cells markedly increased in the HKLs after stimulation (Figure 5C,D). Isotype antibody (mouse IgG1-FITC) was used as control to exclude false positive cells (Appendix A).

### 3.5. Analysis of Promoter Activity of il21

To further analyze the regulatory mechanisms of *il21* expression, we constructed 3 plasmids containing the promoter regions of *il21* gene (Figure 6A). *Il21* pro-1 contained regions 2 kb upstream of the translation initiation codon as it harbored several putative transcriptional factor binding motifs including NF-κB, Sp-1, C/EBPα, CREB, c-Rel, AP-1 and IRF-1 (Appendix A). Figure 6B shows that both *il21* pro-1 and *il21* pro-2 can be activated after stimulation with LPS for 12 h and inactivated *A. hydrophila* (*p* < 0.05 or *p* < 0.01). Similarly, *il21* pro-1 and *il21* pro-2 were activated at 24 h post stimulation (*p* < 0.05 or *p* < 0.01). Furthermore, dose-dependent effects of activation by LPS and inactivated *A. hydrophila* were apparent (*p* < 0.05 or *p* < 0.01) (Figure 6D,E). Conversely, *il21* pro-3 did not respond to either of the stimulants (Figure 6B,C). Taken together, these data indicate that the regulatory elements in the *il21* promoter activated by LPS or inactivated *A. hydrophila* were located in the region between −2000 bp to −433 bp.

### 3.6. Biotesting of Bioactivities of rIL-21 Protein

SDS-PAGE analysis revealed that rIL-21 was approx. 15 kDa (Figure 7A) and was verified by Western blotting using GC9-IL21 mAb (Figure 7B). The rIL-21 protein was then used to stimulate HKLs. Figure 8A–C shows that *β-defensin 1*, *β-defensin 2* and *β-defensin 3* were induced by rIL-21 at the doses of 50 and 100 ng/mL (*p* < 0.01) and that *c-lysozyme* was upregulated by 100 ng/mL rIL-21 (*p* < 0.05) (Figure 8D). However, the *g-lysozyme* expression was unaltered (Figure 8E). In addition, stimulation with 100 ng/mL rIL-21 resulted in upregulation of *il21* (*p* < 0.05), *il21r* (*p* < 0.05) and *il10* (*p* < 0.01) (Figure 8F–H) but downregulation of *tnfα* (*p* < 0.05) (Figure 8J). The expression of *il1β* (Figure 8I) was not affected. Downregulation of *nfκb-p52* and *nfκb-p65* was also detected in the HKLs treated with 50 and 100 ng/mL of rIL-21 (*p* < 0.05 or *p* < 0.01) (Figure 8K,L). We also found that the luciferase activity of the NF-κB promoter in the EPC cells was significantly inhibited by 50 and 100 ng/mL rIL-21 treatment (*p* < 0.05 or *p* < 0.01) (Figure 8M). These results indicate that IL-21 was effective in activating genes involved in antibacterial defense and inhibit NF-κB signaling pathway in fish.

## 4. Discussion

IL-21 is one of the key cytokines mediating innate and adaptive immune responses. Previous studies have shown that fish *il21* genes are expressed mainly in the head kidney, spleen and mucosal tissues including gills and intestines, and are inducible in response to immune stimuli and bacterial infections [14,15,17,18,20,21,22]. In rainbow trout, induction of *il21* expression was detected in the spleen and head kidney of fish infected with *Y. ruckeri* and viral hemorrhagic septicemia virus [17]. Similarly, it was upregulated in tissues of Japanese flounder infected with *E. tarda* [18] and snakehead infected with *A. schubertii* or *N. seriolae* [22]. Consistent with these observations, we found that both the transcript and protein levels of IL-21 were elevated in the kidney, spleen, PBLs, intestine and gills after infection with *A. hydrophila* (Figure 2 and Figure 3). In addition, we and others have shown that stimulation of the primary HKLs of fish with inactivated *A. hydrophila* and LPS significantly induced the transcript expression of *il21* (Figure 5) [14,19,20]. Taken together, these observations indicate that *il21* is activated during bacterial infection.

We determined the biological activities of grass carp IL-21 (Figure 8). Akin to mammalian homologs, fish IL-21s exhibited anti-inflammatory effects and induced the expression of *il10*, a key negative regulator of inflammation [17,19,21,43]. In rainbow trout and grass carp, stimulation with IL-21 significantly increased *il10* expression in the primary HKLs [17,19]. Intraperitoneal injection of IL-21 protein could mitigate the pathological damage of the head kidney and intestine caused by *A. hydrophila* infection [19]. In agreement with these findings, we found that IL-21 induced the expression of *il10* in grass carp HKLs in a dose-dependent manner (Figure 8) but unexpectedly, had no effects on the expression of *il1β*. Interestingly, IL-21 decreased the expression of two important members of the NF-κB family (*nfκb-p52* and *nfκb-p65*). These results suggest that IL-21 exerted anti-inflammatory functions by inhibiting the NF-κB pathway in fish. In support of this notion, it has been shown that mouse IL-21 suppresses NF-κB signaling induced by LPS in peritoneal macrophages [9].

The present study demonstrates that IL-21 activates antimicrobial genes in grass carp. Lysozyme is an important component of the innate immune system and plays a key role in the first line of defense against bacterial infection. In teleost, two types of lysozymes, termed goose-type lysozyme (*g-lysozyme*) and chicken-type lysozyme (*c-lysozyme*), have been identified [44]. In the present study, the expression of *c-lysozyme* but not *g-lysozyme* was increased in the HKLs after stimulation with IL-21 (Figure 8). Other antimicrobial genes, such as *β-defensin 1*, *β-defensin 2* and *β-defensin 3,* were also activated. β-defensins are an important group of antimicrobial peptides which have direct killing activities against bacterial pathogens [36]. Interestingly, in Japanese flounder, IL-21 showed inhibitory effects on the growth of *Streptococcus iniae* [18]. Collectively, these findings highlight the importance of IL-21 in mounting immune defense against bacterial infections.

Teleosts lack Peyer’s patches and mesenteric lymph nodes but instead possess gut associated lymphoid tissues (GALTs) where T lymphocytes and other immune cells reside for interaction with each other [45]. Grass carp are herbivorous fish and do not have a stomach. Instead, they have a widening esophagus and an unusually long intestine. Intestinal health is of great importance to growth performance [41,46]. It has been reported that infection with bacterial pathogens such as *A. hydrophila* causes acute enteritis, leading to high mortalities [29]. On the other hand, fish suffering from chronic enteritis often display a low growth rate. To date, mechanisms on the pathogenesis of enteritis have rarely been investigated in fish. We have shown here that, in the *A. hydrophila* infected fish, the transcript levels of pro-inflammatory cytokines such as *il1β* and *tnfα* were upregulated in the intestine, and immune cells infiltrated into the mucosa and submucosa (Figure 1 and Figure 3). Interestingly, along with *il1β* and *tnfα*, *il21* was also induced. IL-21 has pleiotropic functions and is generally considered an anti-inflammatory cytokine. In mammals, IL-21 plays key roles in the development of inflammatory diseases and intestinal inflammation [47,48,49] and significantly ameliorates DSS-induced intestinal inflammation. It is considered a novel and promising therapeutic target for treatment of patients suffering with inflammatory bowel disease [50,51]. It is likely that a high abundance of IL-21 in the gills and intestine could be beneficial to the body to control excessive inflammation during enteritis caused by *A. hydrophila*. On the other hand, we also observed moderate levels of IL-21 in the gills and intestine of healthy fish in our previous study [21]. This suggests that IL-21 plays an important role in the maintenance of the immune homeostasis of mucosal tissues in fish.

In mammals, IL-21 is produced by activated CD4^+^ T cells and NKT cells [1,2,3]. In rainbow trout and grass carp, the transcripts of *il21* were detected in the HKLs and splenocytes following stimulation with PHA and PMA [17,21], both of which are known T cell stimuli, suggesting that it could be produced by T cells in fish. Using a monoclonal antibody against CD3γ/δ, in the present study, we have shown for the first time that IL-21 is produced by the CD3γ/δ^+^ T cells. The IL-21 producing CD3γ/δ^+^ T cells were found mainly in the submucosa and were induced in the anterior, middle and posterior intestine after *A. hydrophila* infection (Figure 4). It is envisaged that these IL-21-producing CD3γ/δ^+^ T cells could play important roles in the maintenance of immune homeostasis in fish and the control of excessive inflammation during infections.

In summary, we have shown that grass carp *il21* was induced during infection with *A. hydrophila*. Recombinant IL-21 protein was effective in upregulating antibacterial genes, including *β-defensins* and *lysozyme*. In addition, the IL-21-expressing CD3γ/δ T cells were detected in tissues and markedly increased in the anterior, middle and posterior intestine after infection with *A. hydrophila*. Our results suggest that IL-21 plays an important role in regulating intestinal inflammation induced by bacterial infection (Figure 9).

## 5. Conclusions

In conclusion, this study provides evidence that IL-21 is induced and markedly accumulated in the CD3γ/δ T cells of the anterior, middle and posterior intestine of grass carp during *A. hydrophila* infection. Moreover, the recombinant IL-21 protein was effective in elevating the expression of antibacterial genes and inhibiting the NF-κB signaling pathway. The results demonstrate that IL-21 is involved in regulating the intestinal inflammation induced by bacterial infection in grass carp.

## Figures and Tables

**Figure 1 cells-12-02276-f001:**
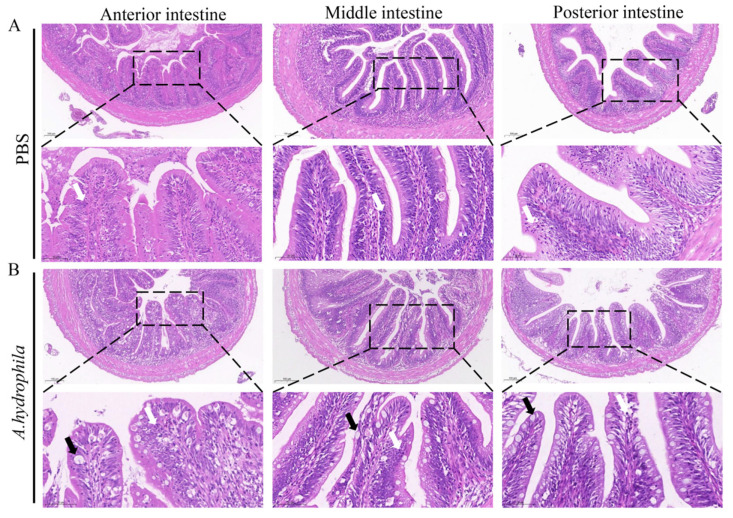
Infection of *A. hydrophila* triggers infiltration of leukocytes in the intestine. (**A**,**B**) Histology of the anterior, middle and posterior intestine of grass carp injected with PBS (control) or *A. hydrophila*. Fish were i.p. injected with 100 μL of *A. hydrophila* (1 × 10^7^ CFU/mL) or PBS. At 72 h, intestine segments were sampled and fixed for histological staining with hematoxylin and eosin. Scale bars = 100 μm. Scale bars of enlarged images are 50 μm. Black and white arrows indicate goblet cells and leucocytes, respectively.

**Figure 2 cells-12-02276-f002:**
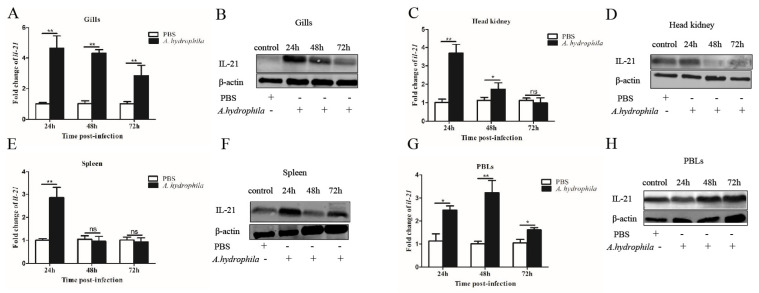
The IL−21 is induced in tissues by *A. hydrophila*. Grass carp were i.p. injected with 100 μL of *A. hydrophila* (1 × 10^7^ CFU/mL) or PBS (control). At 24, 48 and 72 h, mRNA expression levels of *il21* in the gills (**A**), kidney (**C**), spleen (**E**) and PBLs (**G**) were analyzed by qRT−PCR. The data are shown as means ± SE (N = 4). Significant differences are indicated by * *p* < 0.05 and ** *p* < 0.01. ns, no significant difference. Presence of IL−21 protein in the gills (**B**), kidney (**D**), spleen (**F**) and PBLs (**H**) was analyzed by Western blotting.

**Figure 3 cells-12-02276-f003:**
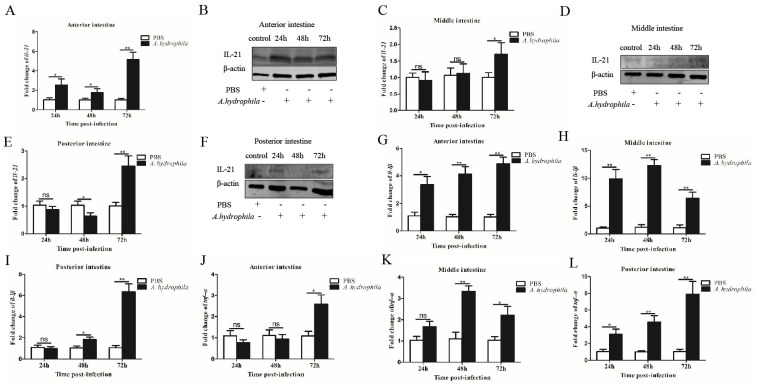
Intestinal IL−21 was induced by *A. hydrophila*. Grass carp were i.p. injected with 100 μL of *A. hydrophila* (1 × 10^7^ CFU/mL) or PBS (control). At 24, 48 and 72 h, the mRNA and protein levels of IL−21 in the anterior intestine (**A**,**B**), middle intestine (**C**,**D**) and posterior intestine (**E**,**F**) were analyzed by qRT−PCR and Western blotting, respectively. The mRNA levels of *il1β* and *tnfα* in the anterior intestine (**G**,**J**), middle intestine (**H**,**K**) and posterior intestine (**I**,**L**) were analyzed by qRT−PCR. The qRT−PCR data are shown as means ± SE (N = 4). Significant differences are indicated by * *p* < 0.05 and ** *p* < 0.01. ns, no significant difference.

**Figure 4 cells-12-02276-f004:**
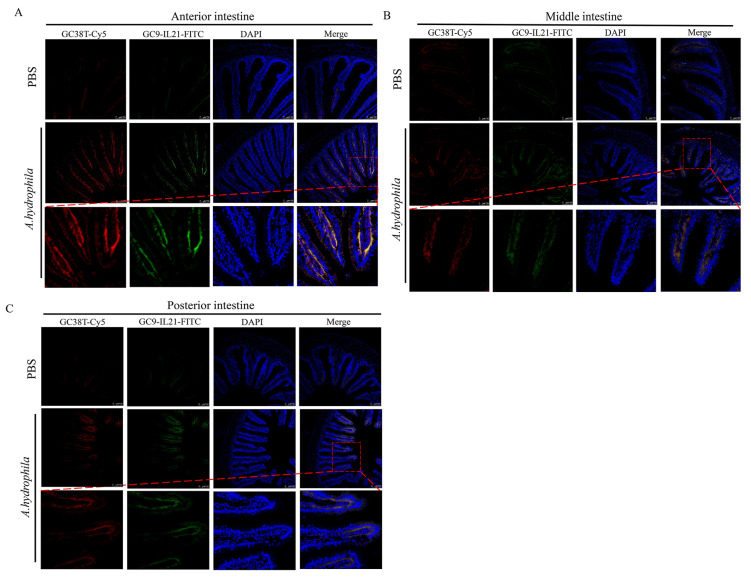
The CD3γ/δ^+^ cells produce IL-21 in intestine. Grass carp were i.p. injected with 100 μL of *A. hydrophila* (1 × 10^7^ CFU/mL) or PBS. After 72 h, anterior (**A**), middle (**B**) and posterior (**C**) intestine were fixed and analyzed by fluorescent microscopy. The IL-21^+^ cells (green) and CD3γ/δ^+^ cells (red) are shown. Nucleus (blue) was counterstained with DAPI.

**Figure 5 cells-12-02276-f005:**
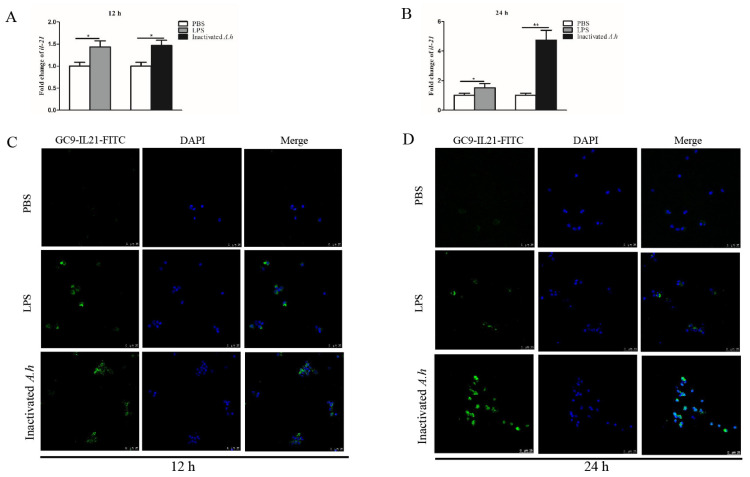
Expression of IL-21 is upregulated in the primary HKLs by LPS and inactivated *A. hydrophila*. Grass carp HKLs were stimulated with PBS, LPS or inactivated *A. hydrophila* (inactivated *A.h*) for 12 and 24 h. The mRNA expression levels were analyzed by qRT-PCR (**A**,**B**) and IL-21^+^ cells by confocal microscopy (**C**,**D**). The IL-21 expressing cells are shown as green. Nucleus (blue) was counterstained with DAPI. Data are shown as means ± SE (N = 4). Significant differences are indicated by * *p* < 0.05 and ** *p* < 0.01. ns, no significant difference.

**Figure 6 cells-12-02276-f006:**
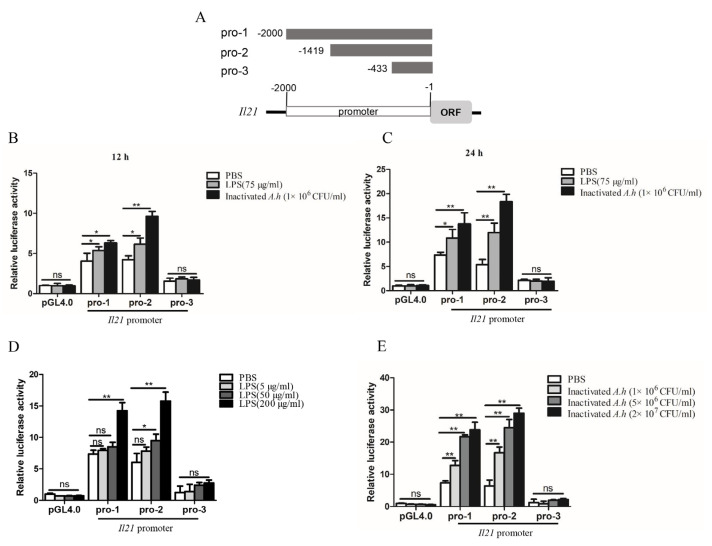
Analysis of the promoter activity of *il21* in the EPC cells. (**A**) Schematic description of *il21* promoter constructs. EPC cells were cultured in 24-well plates overnight and transfected with 50 ng pRL−TK plus 500 ng pGL4.0/pro−1/pro−2/pro−3 (pro−1, pro−2 and pro−3 represent promoters 1, 2 and 3 of *il21*, respectively). At 24 h, cells were stimulated with LPS, inactivated *A. hydrophila* (inactivated *A.h*) or PBS for 12 h (**B**) and 24 h (**C**), or cells were stimulated with different doses of LPS (**D**) or heat inactivated *A. hydrophila* (**E**) for 24 h. Luciferase reporter activity assays were performed. Data are shown as means ± SE (N = 4). Significant differences are indicated by * *p* < 0.05 and ** *p* < 0.01. ns, no significant difference.

**Figure 7 cells-12-02276-f007:**
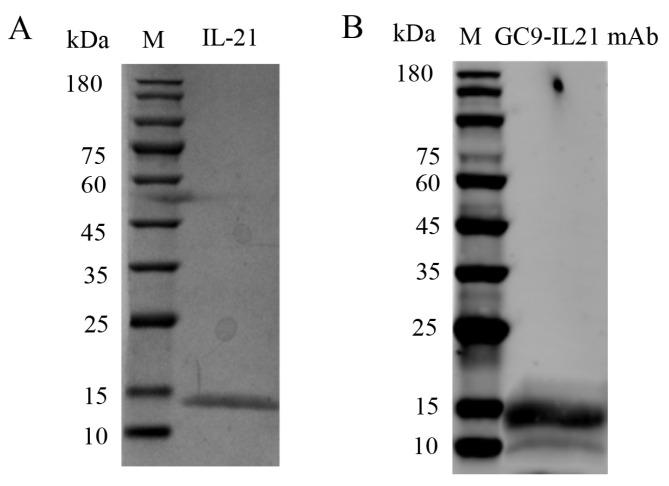
Purified rIL−21 expressed in the CHO−S cells. Purified rIL−21 was analyzed by SDS−PAGE (**A**) and Western blotting (**B**) using GC9−IL21 mAb.

**Figure 8 cells-12-02276-f008:**
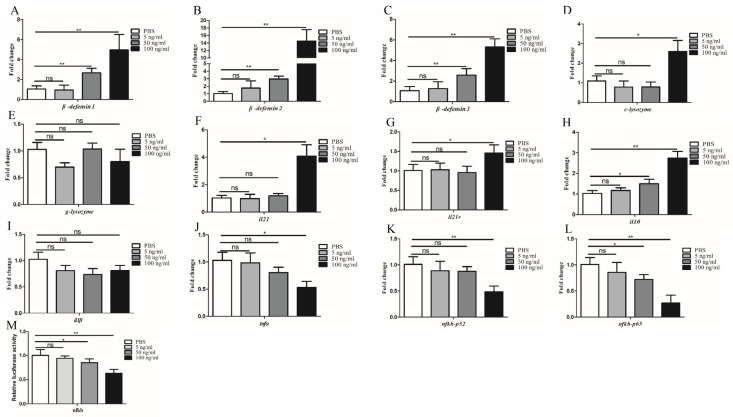
IL-21 activates antimicrobial genes in the HKLs. Grass carp HKLs were stimulated with rIL−21 (5, 50, and 100 ng/mL) or PBS (control). After 24 h, the cells were collected for qRT−PCR analysis of the expression of *β*−*defensin 1* (**A**), *β*−*defensin 2* (**B**), *β*−*defensin 3* (**C**), *c*−*lysozyme* (**D**), *g*−*lysozyme* (**E**), *il21* (**F**), *il21r* (**G**), *il10* (**H**), *il1β* (**I**), *tnfα* (**J**), *nfκb*−*p52* (**K**) and *nfκb*−*p65* (**L**). (**M**) Luciferase activity of the NF−κB promoter in the EPC cells after stimulation with rIL-21 (5, 50 and 100 ng/mL) or PBS (control) for 24 h. Data are shown as means ± SE (N = 4). Significant differences are indicated by * *p* < 0.05 and ** *p* < 0.01. ns, no significant difference.

**Figure 9 cells-12-02276-f009:**
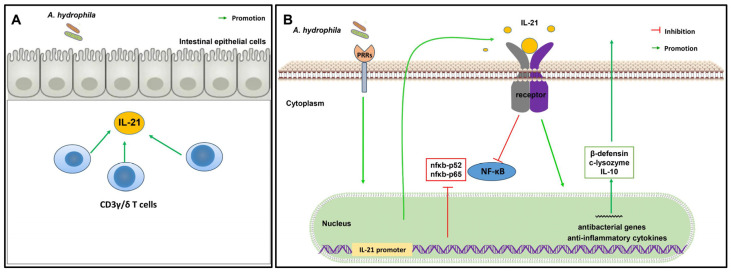
Graphical summary of the expression and biological functions of IL-21 in grass carp during infection with *A. hydrophila*. (**A**) In the intestine of grass carp, *il21* is expressed by the CD3γ/δ T cells after *A. hydrophila* infection. (**B**) The promoter of *il21* is activated by *A. hydrophila,* and *il-21* modulates intestinal inflammation through upregulation of the expression of antibacterial genes, anti-inflammatory cytokines and inhibition of the NF-κB signaling pathway.

## Data Availability

The data are available from J. Zou upon request.

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
