# Peer review of "The Biological Functions and Intestinal Inflammation Regulation of IL-21 in Grass Carp (Ctenopharyngodon idella) during Infection with Aeromonas hydrophila"

_cells, 2023, doi:10.3390/cells12182276_

Round 1

Reviewer 1 Report

This is a well-executed comprehensive study of IL-21 expression in carp during Aeromonas hydrophila infections. It addresses both tissue and cellular sources in vivo and potential mechanisms in vitro. I only have one minor comment. Gene names typically do not have hyphens. Please correct.

Author Response

This is a well-executed comprehensive study of IL-21 expression in carp during Aeromonas hydrophila infections. It addresses both tissue and cellular sources in vivo and potential mechanisms in vitro. I only have one minor comment. Gene names typically do not have hyphens. Please correct.

Response: We thank the reviewer for the constructive comments and for giving the opportunity to revise the manuscript. As suggested, we've deleted hyphens for the gene names.

Reviewer 2 Report

Reviewers' Comments to Authors:

The manuscript entitled The IL-21+/CD3γ/δ+ T cells regulate intestinal inflammation in fish during infection with Aeromonas hydrophila, described by Yuan et al., demonstrates scientific attempts to narrate the biological function of IL-21, a pleiotropic cytokine in grass carp using variously research tools.

Based on scientific consideration, the manuscript contains interesting findings and can be distributed in fish immunology, which is important for understanding crucial information applicable to further initiating prevention methods. All in all, the manuscript is well-organized, but there remain some inconvenient parts, such as trivial errors, that the authors should improve to reach acceptable levels to publish in high-standard journals. The major and minor concerns are as follows;  

Title

It should be properly modified to such “The biological functions and intestinal inflammation regulation of IL-21 in fish grass carp (Ctenopharyngodon idella) during infection with Aeromonas hydrophila. Since the information in the manuscript did not frankly describe the key contents of IL-21+/CD3γ/δ+ T cells regulation in intestinal inflammation.

Abstract

Based on the above recommendation, the information in the abstract should be properly revised.

Introduction

Line 63 and 65. Full information on PHA, PMA, LPS, and poly (I:C) should be indicated.

Line 66. Gene status “il-21” and other places of the other gene should be italicized throughout the manuscript; please carefully check.

Line 76. Please correct “fresh water” to “freshwater” throughout.

Line 79-80. Please revise the following awkward sentence; “On the other hand, intestine is constantly exposed to exogenous microbes, making it vulnerable to infections (25).” Please create a better flow based on the information in the earlier sentences.

Materials and Methods

2.1. Experimental fish, cells and bacteria

Line 103. Unitalicized “Epithelioma papulosum cyprini”.

Line 105. Correct “Penicillin-Streptomycin Solution (Pen/Strep)” to “penicillin-streptomycin solution (Pen/Strep)”.  

Line 106. Please correct “CO2” to “CO2” and everywhere else throughout the manuscript.

Line 109. Please correct “(180 rpm/min)” to “(180 rpm)” and “until OD600” to “until absorbance at optical density (OD) 600 nm” throughout the manuscript.

2.3. Production and bioactivity testing of recombinant IL-21

Line 132. Correct “6x106”.

Line 140. Correct “SDS-PAGE and Western blotting” to “SDS-PAGE and Western blotting, which were described below”

2.4. Bacterial challenge

Line 152. Clarify PBLs: Is it “peripheral blood lymphocytes” or “peripheral blood leukocytes”??

2.5. Expression analysis of IL-21 in the primary HKLs after stimulation

- Verify “after stimulation”.

- Correct “OD600”.

2.6. Luciferase promoter reporter assay

Line 174. Correct by italicizing “A. hydrophila” and other species names of other organisms throughout the manuscript.

2.7. RNA extraction, cDNA synthesis, and qRT-PCR

The conditions of qRT-PCR analysis should be clearly described.

2.8. Western blotting

Line 190. Correct “western”.

2.10. Hematoxylin-eosin (HE) staining of grass carp intestines

Correct “HE” to “H&E”.

2.11. Statistical analysis

One of the most serious contents was falling onto this part. The authors cannot thoroughly apply ANOVA and DMRT mean comparisons for statistically analyzing the experiments' results, which have two treatments. ANOVA and DMPRT can be properly used only in Figure 7.

Therefore, the results in “Figures 2, 3, 5, and 5” must be reanalyzed with factorial in CRD with 2 factors, concentrations, and response times. Furthermore, the interaction of these factors must be properly analyzed.      

Results

Therefore, based on the improper data analysis discussed above, descriptions in this part and “Discussion” must be entirely revised.

3.4. Expression analysis of IL-21 in the primary HKLs after stimulation

- Verify “after stimulation”.

Line 316. Clarify “bacterial stimulation” since the content is not just bacterial induction.

Line 322. Clarify “Isotype antibodies (IgG1-FITC)”.

3.5. Analysis of promoter activity of il-21

Figure 6. The x-axis should be properly labeled.

Line 375. Correct “+” to “±” and the other places throughout.

Discussion

Based on the improper data analysis discussed above, descriptions in this part must be entirely revised.

References

There are many errors, wrong or inconsistent reference formats, punctuation errors, typos, journal name errors, and species names that were observed throughout, not neat. These included references 9, 10, 17, 24, 26, 30, 31, 33, 35, 36, 41, 43, 47-49.

And please carefully check the format of MDPI journals.

It's fine.

Author Response

The manuscript entitled “The IL-21+/CD3γ/δ+ T cells regulate intestinal inflammation in fish during infection with Aeromonas hydrophila”, described by Yuan et al., demonstrates scientific attempts to narrate the biological function of IL-21, a pleiotropic cytokine in grass carp using variously research tools.

Based on scientific consideration, the manuscript contains interesting findings and can be distributed in fish immunology, which is important for understanding crucial information applicable to further initiating prevention methods. All in all, the manuscript is well-organized, but there remain some inconvenient parts, such as trivial errors, that the authors should improve to reach acceptable levels to publish in high-standard journals. The major and minor concerns are as follows;

Response: We thank reviewer for giving us the opportunity to revise the manuscript.

Comment 1: Title

It should be properly modified to such “The biological functions and intestinal inflammation regulation of IL-21 in fish grass carp (Ctenopharyngodon idella) during infection with Aeromonas hydrophila”. Since the information in the manuscript did not frankly describe the key contents of IL-21+/CD3γ/δ+ T cells regulation in intestinal inflammation.

Response: The title is now changed to “The biological functions and intestinal inflammation regulation of IL-21 in grass carp (Ctenopharyngodon idella) during infection with Aeromonas hydrophila”.

Comment 2: Abstract

Based on the above recommendation, the information in the abstract should be properly revised.

Response: As advised, we have carefully checked and corrected the grammar errors and typos in the abstract and text.

Comment 3: Introduction

 Line 63 and 65. Full information on PHA, PMA, LPS, and poly (I:C) should be indicated.

Response: Full names are provided: phytohemagglutinin (PHA), phorbol myristate acetate (PMA), lipopolysaccride (LPS) and polyinosinic-polycytidylic acid (poly (I:C)).

Comment 4 Line 66. Gene status “il-21” and other places of the other gene should be italicized throughout the manuscript; please carefully check.

Response: Hyphens are deleted (see response to reviewer 1) in the gene abbreviations and gene names are in Italics: e.g. il21.

Comment 5 Line 76. Please correct “fresh water” to “freshwater” throughout.

Response: Done.

Comment 6 Line 79-80. Please revise the following awkward sentence; “On the other hand, intestine is constantly exposed to exogenous microbes, making it vulnerable to infections (25).” Please create a better flow based on the information in the earlier sentences.

Response: The sentence is rephrased to “The unusual long intestine can be subdivided into anterior, middle and posterior intestine and are pivotal for food digestion and absorption of nutrients [26], and is one of the major mucosal sites for pathogen invasion [27].”

Comment 7: Materials and Methods 2.1. Experimental fish, cells and bacteria

Line 103. Unitalicized “Epithelioma papulosum cyprini”.

Comment 8: Line 105. Correct “Penicillin-Streptomycin Solution (Pen/Strep)” to “penicillin-streptomycin solution (Pen/Strep)”.  

Comment 9: Line 106. Please correct “CO2” to “CO2” and everywhere else throughout the manuscript.

Comment 10: Line 109. Please correct “(180 rpm/min)” to “(180 rpm)” and “until OD600” to “until absorbance at optical density (OD) 600 nm” throughout the manuscript.

Comment 11: 2.3. Production and bioactivity testing of recombinant IL-21

Line 132. Correct “6x106”.

Comment 12: Line 140. Correct “SDS-PAGE and Western blotting” to “SDS-PAGE and Western blotting, which were described below”

Response to Comments 7-12: All done.

Comment 13: 2.4. Bacterial challenge

Line 152. Clarify PBLs: Is it “peripheral blood lymphocytes” or “peripheral blood leukocytes”??

Response: This is defined in the text: peripheral blood leukocytes (PBLs).

Comment 14: 2.5. Expression analysis of IL-21 in the primary HKLs after stimulation

- Verify “after stimulation”.

- Correct “OD600”.

Response: Revised:

“after stimulation with inactivated A. hydrophila and LPS”

“when the absorbance at optical density (OD) 600 nm of A. hydrophila culture reached 0.6,…”

Comment 15: 2.6. Luciferase promoter reporter assay

Line 174. Correct by italicizing “A. hydrophila” and other species names of other organisms throughout the manuscript.

Response: Done.

Comment 16: 2.7. RNA extraction, cDNA synthesis, and qRT-PCR

The conditions of qRT-PCR analysis should be clearly described.

Response: Relevant information is added: qRT-PCR was performed by using the conditions including: 1 cycle of 95 °C for 30 s, 40 cycles of 95 °C for 5 s, 62 °C for 30 s, 72 °C for 10 s, followed by 1 cycle of 95 °C for 10 s, 65 °C for 60 s, 97 °C for 1 s. qRT-PCR was set up as follows: 5 μL SYBR® Green PreMix Ex Taq™ II (YEASEN, China), 1 μL cDNA template, 0.2 μL forward primer (10 μM), 0.2 μL reverse primer (10 μM), and 3.6 μL H2O.

Comment 17: 2.8. Western blotting

Line 190. Correct “western”.

Response: Done.

Comment 18: 2.10. Hematoxylin-eosin (HE) staining of grass carp intestines

Correct “HE” to “H&E”.

Response: Done.

Comment 19: 2.11. Statistical analysis

One of the most serious contents was falling onto this part. The authors cannot thoroughly apply ANOVA and DMRT mean comparisons for statistically analyzing the experiments' results, which have two treatments. ANOVA and DMPRT can be properly used only in Figure 7.

Therefore, the results in “Figures 2, 3, 5, and 6” must be reanalyzed with factorial in CRD with 2 factors, concentrations, and response times. Furthermore, the interaction of these factors must be properly analyzed.      

Response: We agreed with reviewer’s comments on statistical analysis of datasets and performed Student’s t-test for the datasets of Figures 2, 3, 5 and 6. We obtained similar statistical results which are now presented in the revised figures. Accordingly, we amended the description for section 2.11. Statistical analysis:

Data were statistically analyzed using the SPSS 22.0 software. The statistical significance of biotesting of rIL-21 protein was calculated using Duncan's multiple comparisons of the means and one-way ANOVA test. The statistical significance of others data between treatment group and control group was determined by Student’s t-test. P < 0.05 and P < 0.01 are considered significantly different.

Results

Therefore, based on the improper data analysis discussed above, descriptions in this part and “Discussion” must be entirely revised.

Response: We re-analyzed the datasets using student t test and obtained similar results described in the manuscript (see revised Figures 2, 3, 5 and 6). We revised discussion.

Comment 20: 3.4. Expression analysis of IL-21 in the primary HKLs after stimulation

- Verify “after stimulation”.

Response: Changed to “after stimulation with LPS and inactivated A. hydrophila”

Comment 21: Line 316. Clarify “bacterial stimulation” since the content is not just bacterial induction.

Response: Changed to “stimulation with LPS and inactivated A. hydrophila”.

Comment 22: Line 322. Clarify “Isotype antibodies (IgG1-FITC)”.

Response: Because the mouse monoclonal antibody against grass carp IL-21 belongs to IgG1 subtype, FITC labeled mouse IgG1 (IgG1-FITC) was used as control for immunostaining to exclude false positive cells that reacted with mouse IgG1. Relevant description is provided in “2.2 Plasmids and reagents”.

Comment 23: 3.5. Analysis of promoter activity of il-21

Figure 6. The x-axis should be properly labeled.

Response: We defined “pro-1, pro-2 and pro-3” in the figure legend: pro-1, pro-2 and pro-3 represent promoters 1, 2 and 3 of il21, respectively.

Comment 24: Line 375. Correct “+” to “±” and the other places throughout.

Response: Done.

Comment 25: Discussion

Based on the improper data analysis discussed above, descriptions in this part must be entirely revised.

Response: See previous response.

Comment 26: References

There are many errors, wrong or inconsistent reference formats, punctuation errors, typos, journal name errors, and species names that were observed throughout, not neat. These included references 9, 10, 17, 24, 26, 30, 31, 33, 35, 36, 41, 43, 47-49.

And please carefully check the format of MDPI journals.

Response: We apologize for the errors and formatted the references.

Reviewer 3 Report

The study by Yuan et al. investigated the crucial role of IL-21 in activating the antibacterial defense and regulating the inflammatory responses caused by infection. The study is interesting. However, these comments should be made and clarified.

The title is inappropriate for the aim of the study. The title should be comprehensive. In addition, the species of fish is not included.

The methodology of the study is not clear in the abstract.-

Please define all abbreviations mentioned for the first time.

The introduction mainly talks about IL21; the authors should provide a paragraph about the immune system in fish.

The sample size for all measures should be provided.

The major concern in this study is the number of fish and replication; only 15 fish for each group. Please clarify that PBS is the control L152.

L262: mention the abbreviation after “peripheral blood lymphocytes.”

L182: please mention these tissues, sample size, and weight.

The order of displaying the results should be the same in the material and methods. This order should be the same in the whole manuscript.

The results in this manuscript begin with the methodology, which is already mentioned in the methods section. Please delete and avoid repetition.

L300: for or after 72hs

p-value should be included in the result text.

Figures 6 and 7 are not clear. Please replace them and split Figure 7 into two figures.

moderate editing is required

Author Response

Comment 1: The title is inappropriate for the aim of the study. The title should be comprehensive. In addition, the species of fish is not included.

Response: Changed to “The biological functions and intestinal inflammation regulation of IL-21 in grass carp (Ctenopharyngodon idella) during infection with Aeromonas hydrophila”.

Comment 2:The methodology of the study is not clear in the abstract.

Response: Thanks for the comment. We have added additional information in the abstract.

Comment 3: Please define all abbreviations mentioned for the first time.

Response: Done. See responses to reviewer 2.

Comment 4: The introduction mainly talks about IL21; the authors should provide a paragraph about the immune system in fish.

Response: As commented by the reviewer, our work focuses on IL-21 which is a cytokine molecule. As suggested, we provide following information in the introduction: “Fish are lower vertebrates and comprise both innate and adaptive immune system [24]. Like mammals, fish possess a complex of cytokine network for coordination of immune responses to infections [25].”

Comment 5: The sample size for all measures should be provided.

Response: Thanks. Relevant information is added in “2.3. Bacterial challenge”.

Comment 6: The major concern in this study is the number of fish and replication; only 15 fish for each group. Please clarify that PBS is the control L152.

Response: Sorry for the inaccurate description. Details are provided in the revised M&M: “Fish were intraperitoneally (i.p.) injected with 100 μl of 1 × 107 CFU/ml A. hydrophila or PBS (control). After 24, 48 and 72 h, peripheral blood leukocytes (PBLs) and immune tissues (25 ± 5 mg) including head kidney, gills, spleen, anterior intestine, middle intestine and posterior intestine were sampled from 4 fish of each group at 24, 48 and 72 h after infection for extraction of total RNA and proteins. In addition, three different intestine segments were sampled at 72 h post-infection and fixed in 4% paraformaldehyde (PFA) for cryosection.”.

Comment 7: L262: mention the abbreviation after “peripheral blood lymphocytes.”

Response: Done.

Comment 8: L182: please mention these tissues, sample size, and weight.

Response: This information is added in “2.3. Bacterial challenge”

Comment 9: The order of displaying the results should be the same in the material and methods. This order should be the same in the whole manuscript.

Response: Done.

Comment 10: The results in this manuscript begin with the methodology, which is already mentioned in the methods section. Please delete and avoid repetition.

Response: We deleted the repetitive description of methodology in the results.

Comment 11: L300: for or after 72hs

Response: Changed to “72 h”.

Comment 12: p-value should be included in the result text.

Response: The p-values are added into the results.

Comment 13: Figures 6 and 7 are not clear. Please replace them and split Figure 7 into two figures.

Response: As advised, Figure 6 is revised and Figure 7 is split into two figures (Figures 7 and 8).

Comment 14: Comments on the Quality of English Language: moderate editing is required

Response: Thanks for the comments. English is carefully checked for errors and edited.

Round 2

Reviewer 2 Report

Reviewers' Comments to Authors:

The manuscript entitled The biological functions and intestinal inflammation regulation of IL-21 in fish grass carp (Ctenopharyngodon idella) during infection with Aeromonas hydrophila, described by Yuan et al., demonstrates scientific attempts to narrate the biological function of IL-21, a pleiotropic cytokine in grass carp using variously research tools.

The current version of the manuscript almost responds to all recommendations raised previously, and the quality of it has significantly improved. The " accept " recommendation will be left after the following errors have been appropriately corrected.

1) Line 64. The gene name of “il-21” or other genes should be consistent with either “il-21” or “IL-21” throughout the manuscript; please carefully check once again.  

Line 86. Please correct “fresh-water” to “freshwater” throughout.

2) 2.11. Statistical analysis

One of the most severe contents still falls into this part.

Please change the following content: “The statistical significance of biotesting of rIL-21 protein was calculated using Duncan's multiple comparisons 301 of the means and one-way ANOVA test” to “The statistical significance of biotesting of rIL-21 protein was analyzed for one-way ANOVA and mean comparisons between tested groups were conducted using Duncan's new multiple range test (DMRT)”.  

3) Figure 6. The x-axis should be labeled appropriately in Fig. 6B, 6C, 6D, and 6E.

4) Figure 7. The x-axis should be labeled appropriately in Fig. 7C – 7O.

4) Figure 8. The x-axis should be labeled appropriately in Fig. 6A – 6M.

References

Some errors, wrong or inconsistent reference formats, punctuation errors, typos, journal name errors, and species names were observed. These included references 9, 10, 17, 26, 28, 33, 43.

Reviewer 3 Report

Thank you for the revision.

No further comments.